# The Added Value of Intraventricular Hemorrhage on the Radiomics Analysis for the Prediction of Hematoma Expansion of Spontaneous Intracerebral Hemorrhage

**DOI:** 10.3390/diagnostics12112755

**Published:** 2022-11-10

**Authors:** Te-Chang Wu, Yan-Lin Liu, Jeon-Hor Chen, Yang Zhang, Tai-Yuan Chen, Ching-Chung Ko, Min-Ying Su

**Affiliations:** 1Department of Medical Imaging, Chi-Mei Medical Center, Tainan 71004, Taiwan; 2Department of Medical Sciences Industry, Chang Jung Christian University, Tainan 71101, Taiwan; 3Department of Radiological Sciences, University of California, Irvine, CA 92521, USA; 4Department of Radiology, E-DA Hospital, I-Shou University, Kaohsiung 84001, Taiwan; 5Department of Radiation Oncology, Rutgers-Cancer Institute of New Jersey, Robert Wood Johnson Medical School, New Brunswick, NJ 08901, USA; 6Graduate Institute of Medical Sciences, Chang Jung Christian University, Tainan 71101, Taiwan; 7Center of General Education, Chia Nan University of Pharmacy and Science, Tainan 71710, Taiwan

**Keywords:** spontaneous intracerebral hemorrhage, intraventricular hemorrhage, hematoma expansion, radiomics, prediction

## Abstract

Background: Among patients undergoing head computed tomography (CT) scans within 3 h of spontaneous intracerebral hemorrhage (sICH), 28% to 38% have hematoma expansion (HE) on follow-up CT. This study aimed to predict HE using radiomics analysis and investigate the impact of intraventricular hemorrhage (IVH) compared with the conventional approach based on intraparenchymal hemorrhage (IPH) alone. Methods: This retrospective study enrolled 127 patients with baseline and follow-up non-contrast CT (NCCT) within 4~72 h of sICH. IPH and IVH were outlined separately for performing radiomics analysis. HE was defined as an absolute hematoma growth > 6 mL or percentage growth > 33% of either IPH (HE_P_) or a combination of IPH and IVH (HE_P+V_) at follow-up. Radiomic features were extracted using PyRadiomics, and then the support vector machine (SVM) was used to build the classification model. For each case, a radiomics score was generated to indicate the probability of HE. Results: There were 57 (44.9%) HE_P_ and 70 (55.1%) non-HE_P_ based on IPH alone, and 58 (45.7%) HE_P+V_ and 69 (54.3%) non-HE_P+V_ based on IPH + IVH. The majority (>94%) of HE patients had poor early outcomes (death or modified Rankin Scale > 3 at discharge). The radiomics model built using baseline IPH to predict HE_P_ (RM_P_) showed 76.4% accuracy and 0.73 area under the ROC curve (AUC). The other model using IPH + IVH to predict HE_P+V_ (RM_P+V_) had higher accuracy (81.9%) with AUC = 0.80, and this model could predict poor outcomes. The sensitivity/specificity of RM_P_ and RM_P+V_ for HE prediction were 71.9%/80.0% and 79.3%/84.1%, respectively. Conclusion: The proposed radiomics approach with additional IVH information can improve the accuracy in prediction of HE, which is associated with poor clinical outcomes. A reliable radiomics model may provide a robust tool to help manage ICH patients and to enroll high-risk ICH cases into anti-expansion or neuroprotection drug trials.

## 1. Introduction

Spontaneous intracerebral hemorrhage (sICH) accounts for about 7–15% of all strokes and carries a mortality rate of about 40%, with half of fatalities occurring within the first two days after an ictus [1,2,3]. The hallmark of sICH is the intraparenchymal hemorrhage (IPH). The high rate of early neurological deterioration after sICH is related in part to active bleeding that may proceed for hours after the symptom onset [4]. Among patients undergoing head CT scans within 3 h of sICH onset, 28% to 38% have hematoma expansion (HE) on follow-up CT scans, with volume greater than one third compared with the hematoma volume on original CT scans [3,4]. HE has also been shown to be an independent predictor of clinical deterioration and poor outcomes [3,5,6,7].

Several radiological predictors for HE on baseline non-contrast CT (NCCT) are proposed, including hematoma volume, shape, hypodensities, and density heterogeneity [8,9,10,11,12,13]. The pattern of heterogeneity can be analyzed using texture features extracted by the radiomics approach, which has been shown capable of capturing various agnostic features to aid in HE prediction [14,15,16,17,18,19]. The radiomics features can be further combined with clinical [19] and radiological variables [16,17] to improve the prediction accuracy.

Except for HE in the brain parenchyma, the presence of intraventricular hemorrhage (IVH) at baseline CT scan has been shown to be associated with mortality in patients with sICH [3,20,21]. In more than 33% of sICH patients, IVH was present at baseline CT scan [22,23,24]. IVH was previously described as one risk factor in the ICH score [20], a clinical grading scale for risk stratification of sICH. Another study reported that 30% to 50% of sICH patients experienced additional IVH [21]. Recently, IVH expansion at follow-up CT has also been identified as a strong predictor of poor clinical outcomes [25]. It was shown that including IVH expansion into the definition of HE improves overall prediction accuracy of the 90-day outcome [24]. Nevertheless, the IVH information has usually been ignored in the conventional radiomics models using texture analysis [14,15,16,17,18,19].

The objective of this study was to investigate the added value of IVH for prediction of HE by using the radiomics analysis. The results obtained by considering IVH with IPH were compared to the conventional approach based on the IPH alone. Two different radiomics analyses were performed: (1) using IPH to predict expansion defined based on IPH; (2) using IPH + IVH to predict expansion defined based on IPH + IVH. The performance of the two radiomics analyses for prediction of HE, and for prediction of poor outcome, were compared.

## 2. Materials and Methods

### 2.1. Study Design and Population

In this retrospective, observational study, patients aged > 18 years at 1st episode of sICH who had undergone a baseline and F/U non-contrast CT (NCCT) scan within an interval of 4–72 h from February 2012 to September 2018 in our hospital were included. Patient data were extracted from the sICH database of the picture archiving and communication system (PACS) to identify eligible patients. In total, 178 patients who met the inclusion criteria were identified. The exclusion criteria were: (1) co-existence of vascular lesions and a brain tumor diagnosed during the same admission (N = 16); (2) pediatric patients < 18 years old (N = 3); (3) patients who underwent brain surgery before follow-up CT (N = 27); (4) patients with primary IVH and equivocal IPH at the periventricular regions (N = 5, two illustrated cases in Appendix A). Thus, the data of 127 patients (89 males, 38 females; mean age 60.5 ± 12.8 years; range 30–94 years) were included in the analysis.

### 2.2. Ethical Considerations

The study protocol was approved by the Institutional Review Board of our hospital. Due to the retrospective nature of the study, the IRB waived the requirement to obtain informed consent from participants.

### 2.3. Clinical Parameters and Clinical Outcomes

Clinical information, including blood pressure (SBP > 180 or <180 mmHg; DBP > 100 or <100 mmHg) [26], bleeding diathesis (INR > 1.5, aPTT ratio > 1.5 or platelet count < 1 × 10^5^/mL) [27], Glasgow Coma Scale (GCS) [20] at admission (13~15 or <13) were collected for clinical model analysis. In-hospital mortality and modified Rankin Scale (mRS) [28] at discharge represented the outcomes. A mRS > 3 at discharge was defined as a poor outcome in this study.

### 2.4. CT Imaging Protocol

Brain CT was acquired using our standard protocol on a 64-slice CT (Definition AS; Siemens Medical Solutions, Forchheim, Germany). The scanning range was from the skull base to the cranial vertex with the following parameters: 120 kVp, 380 mAs, and slice thickness/spacing of 4.8/4.8 mm.

### 2.5. Manual Hematoma Segmentation and HE Definition

The segmentation of the ICH region of interest (ROI) was performed manually, using Image J (National Institutes of Health, Bethesda, MD). The ROI drawing for baseline and F/U CT of each patient was done in one sitting by a neuroradiologist (TCW with 15 years of experience). The intraparenchymal hemorrhage (IPH) and intraventricular hemorrhage (IVH) were outlined separately to form two datasets of intracerebral hemorrhage (ICH): ICH_P_ containing the ROIs of IPH, and ICH_P+V_ containing the ROIs of IPH and IVH. Based on the hematoma volumetric change between baseline and F/U CT studies, HE was defined as an absolute hematoma growth > 6 mL or relative growth of >33% from the baseline ICH_P_ [5,29]. ICH_P+V_ has no consensus definition of expansion, so the same criteria were applied. The baseline ROIs of ICH_P_ and ICH_P+V_ were used to extract radiomics features, followed by feature selection and model building to predict HE.

### 2.6. Feature Extraction and Feature Selection

The radiomics analysis (RA) procedures are illustrated in Figure 1.

For the ICH_P_ or ICH_P+V_ in one patient, all segmented ROIs on different slices were combined to form a 3D lesion mask, and the linear interpolation was utilized to convert the hematoma ROI to be isotropic. A total of 1046 radiomic features were extracted using the PyRadiomics open-source python package, including 2D/3D shape, first-order, Gray Level Co-occurrence Matrix (GLCM), Gray Level Size Zone Matrix (GLSZM), Gray Level Run Length Matrix (GLRLM), Gray Level Dependence Matrix (GLDM). The features were extracted from the original and the filtered images, including wavelet-transformed and Laplacian of Gaussian with a kernel of 1, 2, 3 mm. The bin width was set at 25 to minimize the impact of the noise on the extracted quantitative features.

To select robust features that had a high reproducibility, a second hematoma ROI drawing was performed in 30 randomly selected cases by another neuroradiologist (TYC with 21 years of experience). The extracted features from two separately segmented ROIs were correlated to calculate the intraclass correlation coefficient (ICC). Only features with ICC > 0.8 were considered in the subsequent analysis for feature selection by Gaussian radial basis function of support vector machine (SVM) kernel and model building by the kernel approximation classifiers with SVM kernel.

After the above steps, five features from ICH_P_ were selected for the development of prediction model for hematoma expansion, including two GLCM features (JointAverage and Correlation) and three GLRLM features (LongRunLowGrayLevelEmphasis, GrayLevelVariance and_LowGrayLevelRunEmphasis). Another six features were extracted from ICH_P+V_, including one shape feature (SurfaceVolumeRatio), two GLCM features (JointEntropy and InverseVariance), one GLRLM feature (RunPercentage), one GLDM feature (HighGrayLevelEmphasis) and one GLSZM feature (SizeZoneNonUniformityNormalized).

### 2.7. Model Building and Radiomics Score (RS)

The radiomics models (RM) for classification of HE vs. non-HE were built based on either ICH_P_ (RM_P_) or ICH_P+V_ (RM_P+V_). The kernel approximation classifiers with SVM kernel were applied to perform nonlinear classification of data. In order to derive more accurate estimates of prediction performance, the 10-fold cross-validation was used to prevent overfitting, whereby 90% of cases were randomly selected as the training set and the remaining 10% as the testing set. This procedure was repeated ten times to obtain the average results. Two radiomics scores (RS_P_ & RS_P+V_) were calculated for each case using the models built from ICH_P_ and ICH_P+V_. The prediction threshold for hematoma expansion was set at RS ≥ 0. Once the radiomic score ≥ 0, the patient would be classified as an expander.

### 2.8. Statistical Analysis

Statistical analyses of the clinical parameters were performed using SPSS for Windows (V.24.0, IBM, Chicago, IL, USA). Discrete variables are presented as counts (*n*) and percentages (%), and continuous variables are presented as medians and interquartile ranges (IQR). The chi-square test and Student’s t-test were performed for categorical and continuous data, respectively; *p* values < 0.05 were considered statistically significant. The receiver operating characteristic (ROC) curve was constructed to assess the classification performance, and the sensitivity, specificity and accuracy were calculated.

## 3. Results

### 3.1. Hematoma Expansion Status Defined by IPH (HE_P_)

The clinical parameters, hematoma information and short-term outcomes of all 127 patients are summarized in Table 1. When considering the volume change of IPH, a total of 57 patients (44.9%) met the criteria of hematoma expansion (HE_P_) with an absolute hematoma growth > 6 mL (31 cases) or relative growth of >33% (36 cases). The other 70 patients (55.1%) did not meet the criteria and thus were classified as non-HE_P_. Patients with HE_P_ had a higher proportion of alcohol consumption (36.8% vs. 17.1%, *p* = 0.012) and more bleeding diathesis (21.1% vs. 5.7%, *p* = 0.010). In HE_P_ patients, the hematoma volume change was significantly larger in not only IPH (median 38.6 vs. −0.4 mL) but also IVH (median 5.9 vs. 0.1 mL), and combined IPH and IVH (medium 44.2 vs. −0.4 mL). As for the clinical outcomes, HE_P_ patients had more brain surgery (54.4% vs. 30.0%, *p* = 0.005), higher in-hospital mortality (35.1% vs. 4.3%, *p* < 0.001), and overall poor outcomes with a mRS > 3 at discharge (94.7% vs. 62.9%, *p* < 0.001).

Forty-nine patients (38.6%) had IVH at initial presentation. At follow-up, 72 patients exhibited IVH, among whom 25 patients (25/78; 32.1%) had new IVH (i.e., not initially present at baseline). IVH clot retraction (IVH change < 0 mL) was observed in 21 patients (21/49; 42.9%), with two patients exhibiting full resolution of IVH at follow-up. As for the presence of IVH at the baseline, there was no significant difference between patients with or without HE_P_ (39.7% vs. 37.7%). New IVH (34.5% vs. 7.2%), IVH growth > 1 mL (58.6% vs. 8.7%) and any IVH growth (> 0 mL) (67.2% vs. 20.3%) were significantly associated with HE (*p* < 0.001).

### 3.2. Hematoma Expansion Status Defined by IPH + IVH (HE_P+V_)

When using the same criteria of total volume change of >6 mL or relative growth of >33% to define the expansion of IPH and IVH, 58 patients (45.7%) were HE_P+V_ and 69 patients (54.3%) were non-HE_P+V_. In comparison with HE_P_ classification results, five crossover cases were found. Two patients with HE_P_ were re-classified as non-HE_P+V_ (Figure 2a), and three patients with non-HE_P_ were re-classified as HE_P+V_ (Figure 2b). All five patients had poor outcomes. One died, and four survived the episode and were discharged from the hospital with a mRS of 4 or 5. Compared to non-HE_P+V_, HE_P+V_ had higher in-hospital mortality (32.8% vs. 5.8%, *p* < 0.001), and overall poor outcomes (94.8 vs. 62.3%, *p* < 0.001). The clinical parameters, hematoma information and short-term outcomes of all 127 patients based on ICH_P+V_ are summarized in the Appendix A.

### 3.3. HE Prediction Performance of Two Radiomics Models

Two radiomics models were built using the ICH_P_ and ICH_P+V_ on the baseline NCCT to predict HE. The prediction threshold for hematoma expansion was set at radiomics score (RS) ≥ 0. Comparisons of the prediction performance of these two models are summarized in Table 2, and the ROC curves are shown in Figure 3. The radiomics model using conventional IPH to predict HE_P_, i.e., RM_P_, included 41 true positive (TP), 56 true negative (TN), 14 false positive (FP), and 16 false negative (FN) cases. The accuracy, sensitivity, and specificity were 76.4%, 71.9%, and 80.0%, respectively. In the RM_P+V_ using ICH_P+V_ to predict HE_P+V_, the prediction accuracy was improved to 81.9% with 46 TP, 58 TN, 11 FP, and 12 FN cases. The sensitivity and specificity were also improved to 79.3% and 84.1%, respectively. The area under the ROC curve (AUC) of RM_P+V_ was 0.80 (95% CI: 0.72, 0.87) and the AUC of RM_P_ was 0.73 (95% CI: 0.64, 0.80). Figure 4 shows a case example who was classified as an expander using either HE_P_ or HE_P+V_. The model built using IPH alone (RM_P_) gave a false negative result, while the model based on IPH + IVH (RM_P+V_) gave a true positive result and correctly predicted that this patient was an expander. The Appendix A showed the distribution of value of the radiomics score for expanders and non-expanders in these two models. In RM_P_, the value of RS_P_ for expanders and non-expanders ranged from −1.579 to 1.375 (median 0.268; interquartile range (IQR) 0.003~0.762) and −1.510 to 1.243 (median −0.709; IQR −1.013~−0.209), respectively. In RM_P+V_, the value of RS_P+V_ for expanders and non-expanders ranged from −1.250 to 1.425 (median 0.349; IQR 0.039~0.798) and −1.727 to 1.319 (median −0.754, IQR −1.012~−0.347), respectively.

### 3.4. Radiologic Parameters and Early Outcome of Two Radiomics Models

The comparison of radiologic parameters and early outcomes between the labelled HE and non-HE by the two radiomics models is summarized in Table 3. In these 127 sICH patients, the median hospital stay was 20 days with an interquartile range between 12 days and 29 days. Most patients (79.5%; 101 of 127 patients) had a hospital stay < 1 month. Only 26 patients (20.5%) had a hospital stay longer than 30 days and only one patient had a hospital stay longer than 90 days. The ICH patients with hematoma expansion had a significantly longer hospital stay than those without HE (28.8 days vs. 21.4 days, *p* = 0.034). In both prediction models, the labelled HE had significantly larger hematoma volume changes and a higher possibility of poor functional outcome at discharge as compared to the labelled non-HE. In-hospital mortality was significantly higher in the HE labelled by RM_P+V_ (*p* = 0.003) but was not significantly higher in the HE labelled by RM_P_ (*p* = 0.093). In the RM_P+V_, the onsets to CT interval and CT follow-up interval were shorter in the labelled HE with marginal significance (*p* = 0.068 and 0.087, respectively). These findings were consistent with the results of the original definition of HE (Table 1).

## 4. Discussion

The present study investigated the impact of IVH on the radiomics analysis for HE prediction using a case series of 127 patients with sICH. The hematoma ROIs of IPH alone, and IPH with addition of IVH, were used to build separate models. The prediction performance and clinical outcome correlation of these two radiomics models (RM_P_ & RM_P+V_) were compared. RM_P+V_ developed using hematoma ROIs of both IVH and IPH demonstrated better prediction performance of HE and was significantly associated with in-hospital death. That is, when IVH was considered, RM_P+V_ improved the classification accuracy and AUC compared to that of RM_P_ built using the traditional approach based on IPH alone.

Previously reported predictive indicators for HE included the CT angiography spot sign [27,30,31], NCCT radiological features (density heterogeneity, hypodensities, blend sign, etc.) [8,32,33], and clinical information (GCS, onset to CT interval, warfarin use, etc.) [27,31,33,34]. In recent years, the radiomics approach, which uses texture analysis to capture various agnostic features, has also shown convincing results [14,15,16,17,19]. The least absolute shrinkage and selection operation (LASSO) algorithm was the most applied method for feature selection and model building [14,15,16,17,19], presumably due to its wide availability. The more sophisticated SVM algorithm has been applied as well [15]. The accuracy, sensitivity, and specificity ranged from 0.64 to 0.88, 0.75 to 0.89 and 0.60 to 0.87, respectively, covering a wide range, and were highly dependent on the dataset [14,15,16,17,19]. The present study showed comparable results. The model built using the baseline IPH + IVH achieved an accuracy of 81.9%, with a sensitivity of 79.3% and a specificity of 84.1%. For patients with a high risk of expansion, more aggressive procedures, including immediate surgery, may be considered. Another clinical application of the HE prediction model is to identify subjects who are likely to show HE to participate in anti-expansion trials for sICH [13,32]. A high specificity is preferred for this application. That is, patients who are unlikely to show HE should not be enrolled in order to maximize the power of testing the treatment efficacy by using the smallest number of subjects.

The radiomics features used to construct the radiomics model were generally extracted from shape-based, first-order statistics, and second-order statistics. First-order statistics are used to describe the voxel intensity distribution within ROIs. Second-order statistics describe the spatial relationships between neighboring voxels within the ROIs. In general, second-order features are difficult to evaluate by the human visual system. In this study, five features extracted from ROIs of IPH and six features from ROIs of IPH + IVH were chosen for the development of two radiomics models (RM_P_ and RM_P+V_), respectively. The selected features for the development of two radiomics models are different. It could be attributed to the different ROIs of hematoma for feature extraction. Most of these features were filtered or wavelet-transformed second-order texture features, including GLDM features, GLCM features, GLRLM features, and GLSZM features. Among the five features extracted from ROIs of IPH, two GLRLM features (LongRunLowGrayLevelEmphasis and Correlation) and one GLRLM feature (GrayLevelVariance) had been reported as the selected features for the development of radiomics models for HE prediction [16,35,36]. Among the six features from ROIs of IPH and IVH, however, only the shape feature (SurfaceVolumeRatio) had been reported [35]. These features are usually used to describe the density heterogeneity and hypodensity of hematoma, which are proven to be significant predictors for hematoma expansion of sICH [8,30]. Even though radiomics analysis shows promising results for HE prediction, the segmentation of hematoma is required, which hinders its clinical application in the emergency room. Deep learning has been shown to provide a promising solution to achieve expert-level detection and segmentation of intracranial hemorrhage [37,38,39,40,41]. Precise differentiation of IVH and IPH is challenging, which requires judgement based on the knowledge of neuroanatomy and is usually performed manually by experienced neuroradiologists. It is time consuming and also subject to a high intra- and inter-rater variation. The segmentation performed using CT-based planimetry algorithms has been reported in several studies. Cho et al. applied a fully-automated algorithm to perform hematoma segmentation and found mis-interpretation of IPH and IVH at the interface [38]. For IVH adjacent to massive IPH, it might be interpreted as part of IPH or undefined as a result of the distortion of ventricles. Conversely, IPH adjacent to the fourth ventricle might be misidentified as IVH. In the PREDICT cohort study, semi-automated planimetry was applied to perform the volumetric analyses of the total hematoma (IPH + IVH), and the results showed that the minimum detectable differences (MDD) of total hematoma volume were higher in the patients with larger hematoma volumes [42,43] and in the patients with IVH [42]. However, regarding the total intracerebral hematoma (i.e., IPH + IVH), both the semi-automated [42,43] and fully-automated hematoma segmentation algorithm [41,44] showed reliable results compared to manual segmentation.

We implemented a deep learning algorithm using the U-net architecture [45] to perform automatic segmentation of both IPH and IVH. The preliminary model was applied to test the cases included in the present study and achieved the dice similarity coefficient of 0.838. The examples of the IPH and IVH segmentation with concordant and discordant results are illustrated in the Appendix A. Because IVH is present in one-third of sICH patients, a larger dataset of sICH is necessary for the development of a reliable automatic tool for IPH and IVH segmentation. Considering the high frequency of IVH in the sICH patients, the better performance of RM_P+V_ in HE prediction, and the high fidelity of total ICH segmentation using a deep learning approach, integration of such an automated segmentation algorithm into the HE prediction model would be a reasonable clinical approach.

The initial presence of IVH at baseline CT was not associated with HE in the present study, which was consistent with the results found in the PREDICT study [27] and in a cohort study of the BAT score [32] for prediction of ICH expansion. However, IVH had been demonstrated as a risk factor of HE in a case series of 259 patients with putaminal hemorrhage [46] and in the INTERACT study [34]. Our results showed that the dynamic change of IVH, including new IVH (34.5% vs. 7.2%) and any IVH growth (67.2% vs. 20.3%) were significantly associated with HE in the present study (*p* < 0.001). This finding was also consistent with the results of previous studies [21,24,25]. With regard to the early outcomes of the 127 ICH cases in the present study, IVH at the baseline CT scan was also associated with mortality and poor functional outcomes with crude ORs of 4.2 and 4.1, respectively. Our results also showed that the HE cases predicted by RM_P+V_ were correlated with the in-hospital mortality, but not the HE cases predicted by RM_P_. Considering the impact of IVH on prediction of outcomes and the relationship between the dynamic IVH change and HE, the IVH should be considered in future sICH studies.

The present study has several limitations. First, this was a retrospective design using single-center data with a small sample size. Second, the request for F/U CT scans was at each clinician’s discretion, most likely due to the large baseline ICH and/or worsening symptoms. Consequently, there was a relatively high percentage of patients with HE (57/127; 45%), and poor outcomes for almost all expanders (>94%). Third, there was no external testing dataset that could provide a more realistic estimate of the HE prediction performance of the two proposed radiomics models. Without external validation, we cannot ensure that the proposed radiomics model could be applied across various clinical settings. Therefore, this should be considered as a pilot study mainly for proof of principle, to demonstrate the feasibility of the analysis based on combined IPH + IVH. In future studies, AI software may be applied to automatically segment IPH and IVH, to efficiently process a large number of patients and to evaluate the clinical role of the developed radiomics prediction models.

## 5. Conclusions

Compared with conventional radiomics analysis based on IPH per se, addition of IVH in the radiomics analysis to build a model using combined IPH + IVH improves the prediction of the HE. With the maturing of AI software for segmentation of IPH + IVH, the developed model can be implemented in an emergency setting. A reliable model for the prediction of HE will not only provide a useful tool to aid in better management for ICH patients but will also help to select appropriate patients for enrolling into anti-expansion or neuroprotection drug trials.

## Figures and Tables

**Figure 1 diagnostics-12-02755-f001:**
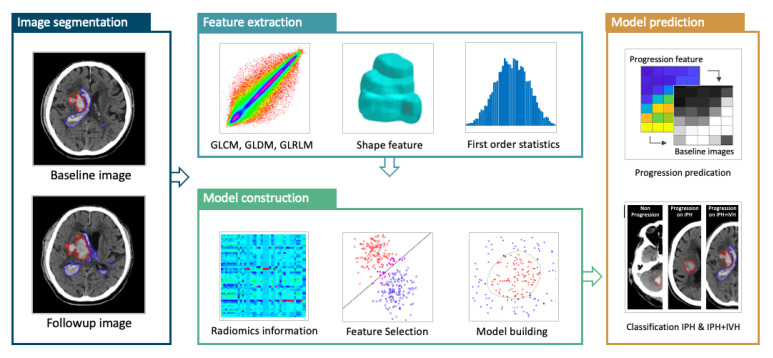
The radiomics analysis flowchart to build the ICH expansion model. The IPH and IVH are segmented by manual tracing of the hematoma at baseline and follow-up CT images. The absolute or percentage volumetric change is calculated to determine whether the patient is an expander or non-expander based on IPH or IPH + IVH. The baseline ROI is used to extract radiomics features, and then the important features are selected using the support vector machine (SVM) algorithm to build the prediction model with the SVM.

**Figure 2 diagnostics-12-02755-f002:**
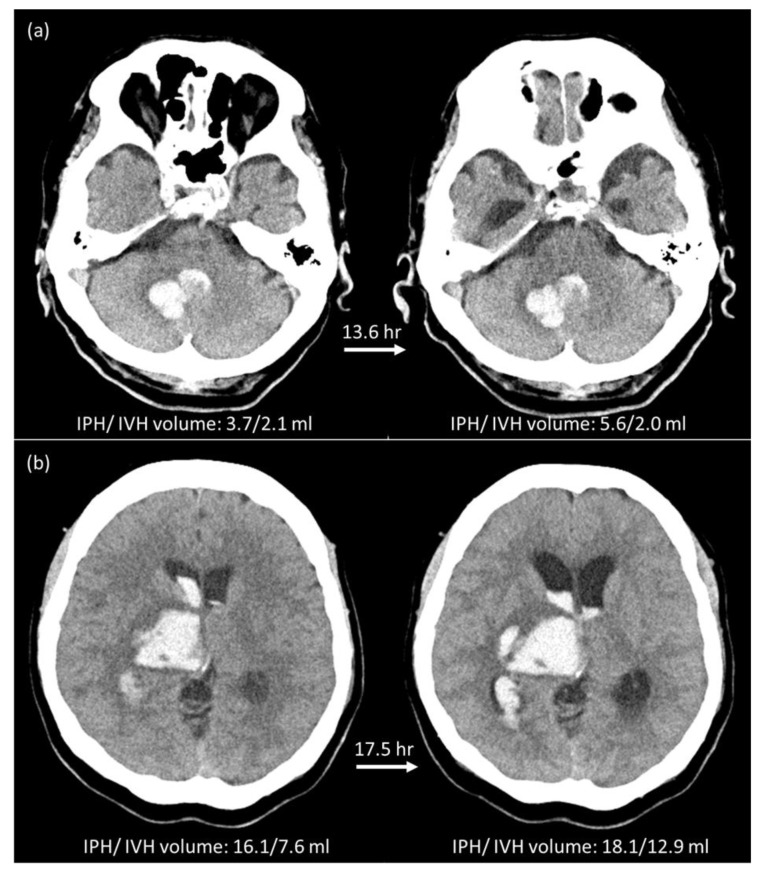
Illustration of two crossover cases. (**a**) A 94-year-old male with right cerebellar hemorrhage was classified as an expander based on IPH (3.7 to 5.6 mL, 51% growth), but was reclassified as a non-expander based on IPH + IVH (5.8 to 7.6 mL, 31% growth < 33% threshold). This patient was discharged on Day-74 after ICH with a mRS of 5. The RM_P_ model showed a true positive result, and the RM_P+V_ showed a true negative result. (**b**) A 52-year-old female with right thalamic hemorrhage was classified as a non-expander based on IPH (16.1 to 18.1 mL) but was re-classified as an expander based on IPH + IVH (23.7 to 31.0 mL, 7.3 mL growth > 6 mL threshold). This patient was discharged on Day-70 after ICH with a mRS of 5. The RM_P_ model showed a true negative result but the RM_P+V_ showed a false negative result.

**Figure 3 diagnostics-12-02755-f003:**
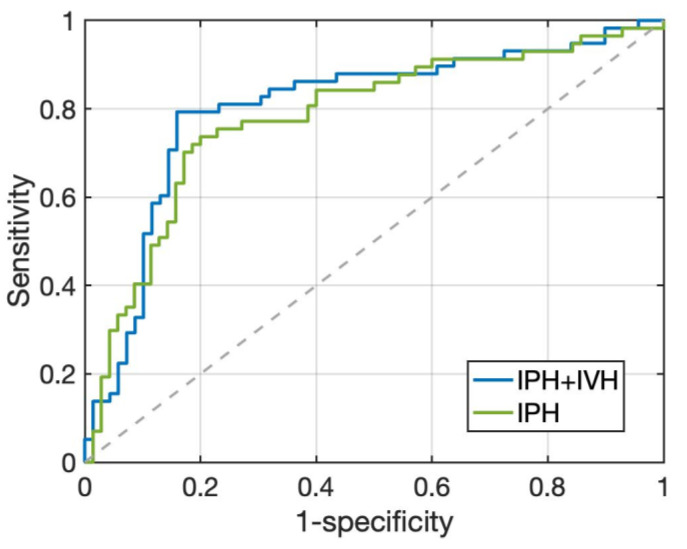
The ROC curves of RM_P_ built using IPH, and RM_P+V_ built using IPH + IVH. The AUC is 0.73 (95% CI: 0.64, 0.80) for RM_P_, and 0.80 (95% CI: 0.72, 0.87) for RM_P+V_.

**Figure 4 diagnostics-12-02755-f004:**
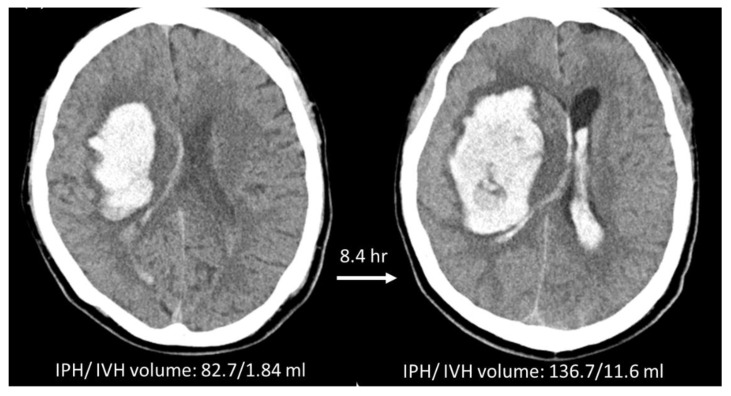
A case illustration. A 51-year-old male with right putaminal hemorrhage, classified as an expander based either on IPH or IPH + IVH criteria. The patient was discharged on Day-5 after ICH with a mRS of 6. The RM_P_ based on ICH_P_ wrongly predicted this case as a non-expander, but the RM_P+V_ based on ICH_P+V_ correctly predicted this case as an expander.

**Table 1 diagnostics-12-02755-t001:** The clinical parameters, hematoma information and short-term outcomes of 127 sICH patients based on ICH_P_.

	Hematoma Expansion Based on ICH_P_		
	Yes (57 Cases)	No (70 Cases)	*p* Value
Sex			0.681
Male/Female	41/16 (72%/28%)	48/22 (69%/31%)	
Age (years)	60.6 (51, 69)	60.4 (51, 67)	0.921
Interval from onset to CT scan (min)	165 (60, 168)	200 (76, 207)	0.379
Interval between CT scans (h)	19.9 (4.6, 24.8)	23.8 (9.9, 35.9)	0.327
Initial IPH volume (mL)	17.2 (7.2,23.4)	22.7 (9.8, 28.8)	0.064
Initial IVH volume (mL)	4.5 (0, 4.4)	1.8 (0, 2)	0.097
Initial IPH + IVH volume (mL)	21.7 (10.6, 28.1)	24.5 (10.5, 34.2)	0.423
IPH volume change (mL)	38.6 (10.6, 63.3)	−0.4 (−1.8, 1.3)	<0.001 *
IVH volume change (mL)	5.6 (0, 9.4)	0.1 (0, 0)	<0.001 *
IPH + IVH volume change (mL)	44.2 (11.9, 72.8)	−0.4 (−2.6, 1.4)	<0.001 *
IVH at baseline CT scan	23 (40.4%)	26 (37.1%)	0.712
DM	15 (26.3%)	20 (28.6%)	0.777
HTN	45 (78.9%)	58 (82.8%)	0.576
Smoking	26 (45.6%)	22 (31.4%)	0.101
Alcohol	21 (36.8%)	12 (17.1%)	0.012 *
Antiplatelet/Anticoagulation	13 (22.8%)	10 (14.3%)	0.215
Bleeding diathesis ^#^	12 (21.1%)	4 (5.7%)	0.010 *
SBP at ER > 180 mmHg	31 (54.4%)	35 (50.0%)	0.623
DBP at ER > 100 mmHg	36 (63.2%)	40 (57.1%)	0.492
GCS 3–12	24 (42.1%)	31 (44.3%)	0.805
Location			0.071
basal ganglia	34 (59.6%)	31 (44.3%)	
thalamus	9 (15.8%)	22 (31.4%)	
lobar	6 (10.5%)	12 (17.1%)	
posterior fossa	8 (14.0%)	5 (7.1%)	
Hospital stay (days)	20 (11, 41.5)	19.5 (12, 26)	0.034 *
In-hospital mortality	20 (35.1%)	3 (4.3%)	<0.001 *
Brain surgery during hospitalization	31 (54.4%)	21 (30.0%)	0.005 *
Poor outcome (mRS > 3 at discharge)	54 (94.7%)	44 (62.9%)	<0.001 *

For continuous variables, median (25%, 75%) values are reported. For number of patients, N (%) are reported. ^#^ INR > 1.5, aPTT > 1.5 or Platelet count < 100,000/uL. * Statistically significant difference (*p* < 0.05).

**Table 2 diagnostics-12-02755-t002:** The prediction performance of two different radiomics analysis models.

	RM_P+V_	RM_p_
Accuracy	81.9% (104/127)	76.4% (97/127)
Sensitivity	79.3% (46/58)	71.9% (41/57)
Specificity	84.1% (58/69)	80.0% (56/70)
False Positive Rate	19.3% (11/57)	25.5% (14/55)
False Negative Rate	17.1% (12/70)	22.2% (16/72)
Positive Predictive Value	80.7% (46/57)	74.5% (41/55)
Negative Predictive Value	82.9% (58/70)	77.8% (56/72)

**Table 3 diagnostics-12-02755-t003:** Comparison of radiologic parameters and clinical outcomes of the labelled patients in different prediction models.

	Labelled Hematoma Expansion		
	Yes	No	*p* Value
RM_P+V_	53 cases	74 cases	
Median interval from onset to CT scan (min)	141 (55, 149)	215 (80, 221)	0.068
Median interval between CT scans (h)	18.1 (5.1, 22.9)	24.9 (9.5, 40.2)	0.087
Median initial IPH volume (mL)	22.5 (9.8, 25.8)	17.6 (6.9, 26.4)	0.103
Median initial IVH volume (mL)	3.5 (0, 4.8)	0 (0, 4.8)	0.737
Median initial IPH + IVH volume (mL)	25.5 (9.8, 26.7)	21.1 (10.9, 30.6)	0.223
Median IPH volume change (mL)	31.3 (4.3, 51.7)	7.0 (−1.2, 2.9)	<0.001
Median IVH volume change (mL)	4.2 (0, 5.0)	1.4 (−0.1, 0.6)	0.029
Median IPH + IVH volume change (mL)	35.5 (4.8, 65.4)	0.9 (−1.3, 5.7)	<0.001
Intraventricular extension	16 (30.2%)	33 (44.6%)	0.100
GCS 3–13	20 (37.7%)	35 (47.3%)	0.284
Hospital stay (days)	19 (11.5, 27)	21 (12, 30.3)	0.747
Brain surgery during hospitalization	26 (49.1%)	26 (35.1%)	0.116
In-hospital mortality	16 (30.2%)	7 (9.5%)	0.003
mRS at discharge > 3	49 (92.5%)	57 (77.0%)	0.021
RM_P_	52 cases	75 cases	
Median interval from onset to CT scan (min)	156 (62, 168)	204 (72, 217)	0.237
Median interval between CT scans (h)	18.6 (4.2, 23.6)	24.4 (8.9, 40.0)	0.145
Median initial IPH volume (mL)	22.5 (10.4, 30.2)	17.7 (7.0, 23.0)	0.108
Median initial IVH volume (mL)	2.9 (0, 2.6)	3.5 (0, 4.1)	0.718
Median initial IPH + IVH volume (mL)	25.4 (10.4, 36.8)	21.2 (10.7, 28.0)	0.235
Median IPH volume change (mL)	32.7 (4.1, 48.6)	6.3 (−1.3, 4.3)	<0.001
Median IVH volume change (mL)	4.6 (0, 7.8)	1.2 (0, 1.0)	0.010
Median IPH + IVH volume change (mL)	37.3 (4.4, 57.5)	7.4 (−1.4, 5.5)	<0.001
Intraventricular extension	20 (38.5%)	29 (38.7%)	0.981
GCS 3–13	20 (38.5%)	35 (46.7%)	0.359
Hospital stay (days)	22 (12.3, 39)	19 (12, 26)	0.087
Brain surgery during hospitalization	28 (53.4%)	24 (32%)	0.014
In-hospital mortality	13 (25%)	10 (13.3%)	0.093
mRS at discharge > 3	48 (92.3%)	58 (77.3%)	0.026

## Data Availability

The complete data are available from the corresponding author on a reasonable request.

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
