# Peer review of "The Added Value of Intraventricular Hemorrhage on the Radiomics Analysis for the Prediction of Hematoma Expansion of Spontaneous Intracerebral Hemorrhage"

_diagnostics, 2022, doi:10.3390/diagnostics12112755_

Round 1

Reviewer 1 Report

Thank you for giving me the opportunity to review the manuscript entitled "The added value of intraventricular hemorrhage on the radiomics analysis for the prediction of hematoma expansion of spontaneous intracerebral hemorrhage" in which the authors evaluate the additional value of intracranial hemorrhage to predict hematoma growth in patients with ICH using a radiomics approach.

The manuscript is well written and structured and addresses an important topic in daily clinical routine. There are some typos and the manuscript should be proof read.

Introduction:

- focused, appropriate length and literature referred to;

Material and Methods:

- well presented, conclusive

Results

- conclusive and sound

Discussion

- appropriate length; conclusion can be derived from the data presented

- appropriate literature referred to

Figures

- good quality

Author Response

Response to Reviewer 1 Comments

Dear respected Editor and Reviewers,

Thank you very much for the positive and encouraging review and comments on our manuscript.

We have discussed all the critical issues/concerns among the coauthors and did our best to do point-to-point response, clarification, and improvement. We believe the clarity and scientific quality of the manuscript has been greatly improved. Thank you.

Review 1

Comments and Suggestions for Authors

Thank you for giving me the opportunity to review the manuscript entitled "The added value of intraventricular hemorrhage on the radiomics analysis for the prediction of hematoma expansion of spontaneous intracerebral hemorrhage" in which the authors evaluate the additional value of intracranial hemorrhage to predict hematoma growth in patients with ICH using a radiomics approach.

Point 1: The manuscript is well written and structured and addresses an important topic in daily clinical routine. There are some typos and the manuscript should be proof read.

Response 1: Thanks for your recommendation. The typos in the manuscript had been corrected in this revision.

Reviewer 2 Report

This study proposed Radiomic-based prediction of hematoma expansion. Two models that use radiomic features from either IPH or a combination of IPH and IVH were investigated.  SVMs were adopted as predictors. This study is interesting, was performed with a scientific and systematic workflow. The manuscript was well-organized and easy to understand. But I have some points that the authors should clarify.

1.     In feature selection, what have the features been finally selected? Could the authors discuss the selected features in the radiology diagnosis aspect?

2.     What was the type of kernel of SVM applied?

3.     What is the range of value of the radiomic score and its meaning?

4.     In the discussion, the authors mentioned that there are automatic segmentations of both IPH and IVH with a high dice similarity coefficient have been proposed. Why did the authors not adopt an automatic segmentation method in this study? If a segmentation method is applied, will it affect the performance of the proposed method?

Round 2

Reviewer 2 Report

Most of my comments were addressed satisfactorily.